# Learning Latent Subspaces in Variational Autoencoders

**Jack Klys, Jake Snell, Richard Zemel**
University of Toronto
Vector Institute
{jackklys,jsnell,zemel}@cs.toronto.edu

## Abstract

Variational autoencoders (VAEs) [10, 20] are widely used deep generative models capable of learning unsupervised latent representations of data. Such representations are often difficult to interpret or control. We consider the problem of unsupervised learning of features correlated to specific labels in a dataset. We propose a VAE-based generative model which we show is capable of extracting features correlated to binary labels in the data and structuring it in a latent subspace which is easy to interpret. Our model, the Conditional Subspace VAE (CSVAE), uses mutual information minimization to learn a low-dimensional latent subspace associated with each label that can easily be inspected and independently manipulated. We demonstrate the utility of the learned representations for attribute manipulation tasks on both the Toronto Face [23] and CelebA [15] datasets.

## 1 Introduction

Deep generative models have recently made large strides in their ability to successfully model complex, high-dimensional data such as images [8], natural language [1], and chemical molecules [6]. Though useful for data generation and feature extraction, these unstructured representations still lack the ease of understanding and exploration that we desire from generative models. For example, the correspondence between any particular dimension of the latent representation and the aspects of the data it is related to is unclear. When a latent feature of interest is labelled in the data, learning a representation which isolates it is possible [11, 21], but doing so in a fully unsupervised way remains a difficult and unsolved task.

Consider instead the following slightly easier problem. Suppose we are given a dataset of $N$ labelled examples $\mathcal{D} = \{(\mathbf{x}_1, y_1), \ldots, (\mathbf{x}_N, y_N)\}$ with each label $y_i \in \{1, \ldots, K\}$, and data belonging to each class $y_i$ has some latent structure (for example, it can be naturally clustered into sub-classes or organized based on class-specific properties). Our goal is to learn a generative model in which this structure can easily be recovered from the learned latent representations. Moreover, we would like our model to allow manipulation of these class-specific properties in any given new data point (given only a single example), or generation of data with any class-specific property in a straightforward way.

We investigate this problem within the framework of variational autoencoders (VAE) [10, 20]. A VAE forms a generative distribution over the data $p_\theta(\mathbf{x}) = \int p(\mathbf{z}) p_\theta(\mathbf{x}|\mathbf{z}) \, d\mathbf{z}$ by introducing a latent variable $\mathbf{z} \in \mathcal{Z}$ and an associated prior $p(\mathbf{z})$. We propose the Conditional Subspace VAE (CSVAE), which learns a latent space $\mathcal{Z} \times \mathcal{W}$ that separates information correlated with the label $y$ into a predefined subspace $\mathcal{W}$. To accomplish this we require that the mutual information between $\mathbf{z}$ and $y$ should be 0, and we give a mathematical derivation of our loss function as a consequence of imposing this condition on a directed graphical model. By setting $\mathcal{W}$ to be low dimensional we can easily analyze the learned representations and the effect of $\mathbf{w}$ on data generation.

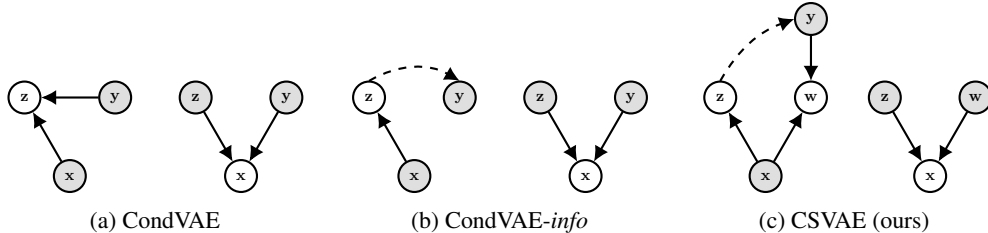

(a) CondVAE        (b) CondVAE-*info*        (c) CSVAE (ours)

Figure 1: The encoder (left) and decoder (right) for each of the baselines and our model. Shaded nodes represent conditioning variables. Dotted arrows represent adversarially trained prediction networks used to minimize mutual information between variables.

The aim of our CSVAE model is twofold:

1. Learn higher-dimensional latent features correlated with binary labels in the data.

2. Represent these features using a subspace that is easy to interpret and manipulate when generating or modifying data.

We demonstrate these capabilities on the Toronto Faces Dataset (TFD) [23] and the CelebA face dataset [15] by comparing it to baseline models including a conditional VAE [11, 21] and a VAE with adversarial information minimization but no latent space factorization [3]. We find through quantitative and qualitative evaluation that the CSVAE is better able to capture intra-class variation and learns a richer yet easily manipulable latent subspace in which attribute style transfer can easily be performed.

## 2 Related Work

There are two main lines of work relevant to our approach as underscored by the dual aims of our model listed in the introduction. The first of these seeks to introduce useful structure into the latent representations of generative models such as Variational Autoencoders (VAEs) [10, 20] and Generative Adversarial Networks (GANs) [7]. The second utilizes trained machine learning models to manipulate and generate data in a controllable way, often in the form of images.

**Incorporating structure into representations.** A common approach is to make use of labels by directly defining them as latent variables in the model [11, 22]. Beyond providing an explicit variable for the labelled feature this yields no other easily interpretable structure, such as discovering features correlated to the labels, as our model does. This is the case also with other methods of structuring latent space which have been explored, such as batching data according to labels [12] or use of a discriminator network in a non-generative model [13]. Though not as relevant to our setting, we note there is also recent work on discovering latent structure in an unsupervised fashion [2, 9].

An important aspect of our model used in structuring the latent space is mutual information minimization between certain latent variables. There are other works which use this idea in various ways. In [3] an adversarial network similar to the one in this paper is used, but minimizes information between the latent space of a VAE and the feature labels (see Section 3.3). In [16] independence between latent variables is enforced by minimizing maximum mean discrepancy, and it is an interesting question what effect their method would have in our model, which we have not pursued here. Other works which utilize adversarial methods in learning latent representations which are not as directly comparable to ours include [4, 5, 17].

**Data manipulation and generation.** There are also several works that specifically consider transferring attributes in images as we do here. The works [26], [24], and [25] all consider this task, in which attributes from a source image are transferred onto a target image. These models can perform attribute transfer between images (e.g. "splice the beard style of image A onto image B"), but only through interpolation between existing images. Once trained our model can modify an attribute of a single given image to any style encoded in the subspace.

## 3 Background

### 3.1 Variational Autoencoder (VAE)

The variational autoencoder (VAE) [10, 20] is a widely-used generative model on top of which our model is built. VAEs are trained to maximize a lower bound on the marginal log-likelihood $\log p_\theta(\mathbf{x})$ over the data by utilizing a learned approximate posterior $q_\phi(\mathbf{z}|\mathbf{x})$:

$$\log p_\theta(\mathbf{x}) \geq \mathbb{E}_{q_\phi(\mathbf{z}|\mathbf{x})}\left[\log p_\theta(\mathbf{x}|\mathbf{z})\right] - D_{KL}\left(q_\phi(\mathbf{z}|\mathbf{x}) \parallel p(\mathbf{z})\right) \tag{1}$$

Once training is complete, the approximate posterior $q_\phi(\mathbf{z}|\mathbf{x})$ functions as an encoder which maps the data $\mathbf{x}$ to a lower dimensional latent representation.

### 3.2 Conditional VAE (CondVAE)

A conditional VAE [11, 21] (CondVAE) is a supervised variant of a VAE which models a labelled dataset. It conditions the latent representation $\mathbf{z}$ on another variable $\mathbf{y}$ representing the labels. The modified objective becomes:

$$\log p_\theta\left(\mathbf{x}|\mathbf{y}\right) \geq \mathbb{E}_{q_\phi(\mathbf{z}|\mathbf{x},\mathbf{y})}\left[\log p_\theta\left(\mathbf{x}|\mathbf{z},\mathbf{y}\right)\right] - D_{KL}\left(q_\phi\left(\mathbf{z}|\mathbf{x},\mathbf{y}\right) \parallel p\left(\mathbf{z}\right)\right) \tag{2}$$

This model provides a method of structuring the latent space. By encoding the data and modifying the variable $\mathbf{y}$ before decoding it is possible to manipulate the data in a controlled way. A diagram showing the encoder and decoder is in Figure 1a.

### 3.3 Conditional VAE with Information Factorization (CondVAE-*info*)

The objective function of the conditional VAE can be augmented by an additional network $r_\psi(\mathbf{z})$ as in [3] which is trained to predict $\mathbf{y}$ from $\mathbf{z}$ while $q_\phi\left(\mathbf{z}|\mathbf{x}\right)$ is trained to minimize the accuracy of $r_\psi$. In addition to the objective function (2) (with $q_\phi\left(\mathbf{z}|\mathbf{x},\mathbf{y}\right)$ replaced with $q_\phi\left(\mathbf{z}|\mathbf{x}\right)$), the model optimizes

$$\max_\phi \min_\psi L(r_\psi(q_\phi\left(\mathbf{z}|\mathbf{x}\right)), \mathbf{y}) \tag{3}$$

where $L$ denotes the cross entropy loss. This removes information correlated with $\mathbf{y}$ from $\mathbf{z}$ but the encoder does not use $\mathbf{y}$ and the generative network $p\left(\mathbf{x}|\mathbf{z},\mathbf{y}\right)$ must use the one-dimensional variable $\mathbf{y}$ to reconstruct the data, which is suboptimal as we demonstrate in our experiments. We denote this model by CondVAE-*info* (diagram in Figure 1b). In the next section we will give a mathematical derivation of the loss (3) as a consequence of a mutual information condition on a probabilistic graphical model.

## 4 Model

### 4.1 Conditional Subspace VAE (CSVAE)

Suppose we are given a dataset $\mathcal{D}$ of elements $(\mathbf{x}, \mathbf{y})$ with $\mathbf{x} \in \mathbb{R}^n$ and $\mathbf{y} \in Y = \{0,1\}^k$ representing $k$ features of $\mathbf{x}$. Let $H = Z \times W = Z \times \prod_{i=1}^k W_i$ denote a probability space which will be the latent space of our model. Our goal is to learn a latent representation of our data which encodes all the information related to feature $i$ labelled by $\mathbf{y}_i$ exactly in the subspace $W_i$.

We will do this by maximizing a form of variational lower bound on the marginal log likelihood of our model, along with minimizing the mutual information between $Z$ and $Y$. We parameterize the joint log-likelihood and decompose it as:

$$\log p_{\theta,\gamma}\left(\mathbf{x}, \mathbf{y}, \mathbf{w}, \mathbf{z}\right) = \log p_\theta\left(\mathbf{x}|\mathbf{w}, \mathbf{z}\right) + \log p\left(\mathbf{z}\right) + \log p_\gamma\left(\mathbf{w}|\mathbf{y}\right) + \log p\left(\mathbf{y}\right) \tag{4}$$

where we are assuming that $Z$ is independent from $W$ and $Y$, and $X \mid W$ is independent from $Y$. Given an approximate posterior $q_\phi\left(\mathbf{z}, \mathbf{w}|\mathbf{x}, \mathbf{y}\right)$ we use Jensen's inequality to obtain the variational lower bound

$$\begin{aligned}
\log p_{\theta,\gamma}\left(\mathbf{x}, \mathbf{y}\right) &= \log \mathbb{E}_{q_\phi(\mathbf{z},\mathbf{w}|\mathbf{x},\mathbf{y})}\left[p_{\theta,\gamma}\left(\mathbf{x}, \mathbf{y}, \mathbf{w}, \mathbf{z}\right)/q_\phi\left(\mathbf{z}, \mathbf{w}|\mathbf{x}, \mathbf{y}\right)\right] \\
&\geq \mathbb{E}_{q_\phi(\mathbf{z},\mathbf{w}|\mathbf{x},\mathbf{y})}\left[\log p_{\theta,\gamma}\left(\mathbf{x}, \mathbf{y}, \mathbf{w}, \mathbf{z}\right)/q_\phi\left(\mathbf{z}, \mathbf{w}|\mathbf{x}, \mathbf{y}\right)\right].
\end{aligned}$$

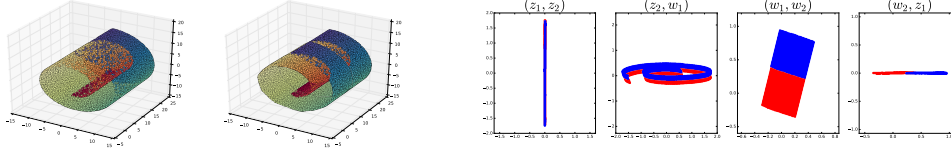

Figure 2: Left: The swiss roll and its reconstruction by CSVAE. Right: Projections onto the axis planes of the latent space of CSVAE trained on the swiss roll, color coded by labels. The data overlaps in $Z$ making it difficult for the model to determine the label of a data point from this projection alone. Conversely the data is separated in $W$ by its label.

Using (4) and taking the negative gives an upper bound on $-\log p_{\theta,\gamma}(\mathbf{x}, \mathbf{y})$ of the form

$$m_1(\mathbf{x}, \mathbf{y}) = -\mathbb{E}_{q_\phi(\mathbf{z}, \mathbf{w}|\mathbf{x}, \mathbf{y})}\left[\log p_\theta(\mathbf{x}|\mathbf{w}, \mathbf{z})\right] + D_{KL}\left(q_\phi(\mathbf{w}|\mathbf{x}, \mathbf{y}) \,\|\, p_\gamma(\mathbf{w}|\mathbf{y})\right)$$
$$+ D_{KL}\left(q_\phi(\mathbf{z}|\mathbf{x}, \mathbf{y}) \,\|\, p(\mathbf{z})\right) - \log p(\mathbf{y}).$$

Thus we obtain the first part of our objective function:

$$\mathcal{M}_1 = \mathbb{E}_{\mathcal{D}(\mathbf{x}, \mathbf{y})}\left[m_1(\mathbf{x}, \mathbf{y})\right] \tag{5}$$

We derived (5) using the assumption that $Z$ is independent from $Y$ but in practice minimizing this objective will not imply that our model will satisfy this condition. Thus we also minimize the mutual information

$$I(Y; Z) = \mathcal{H}(Y) - \mathcal{H}(Y|Z)$$

where $\mathcal{H}(Y|Z)$ is the conditional entropy. Since the prior on $Y$ is fixed this is equivalent to maximizing the conditional entropy

$$\mathcal{H}(Y|Z) = \iint_{Z,Y} p(\mathbf{z}) p(\mathbf{y}|\mathbf{z}) \log p(\mathbf{y}|\mathbf{z}) \, d\mathbf{y} d\mathbf{z}$$
$$= \iiint_{Z,Y,X} p(\mathbf{z}|\mathbf{x}) p(\mathbf{x}) p(\mathbf{y}|\mathbf{z}) \log p(\mathbf{y}|\mathbf{z}) \, d\mathbf{x} d\mathbf{y} d\mathbf{z}.$$

Since the integral over $Z$ is intractable, to approximate this quantity we use approximate posteriors $q_\delta(\mathbf{y}|\mathbf{z})$ and $q_\phi(\mathbf{z}|\mathbf{x})$ and instead average over the empirical data distribution

$$\mathbb{E}_{\mathcal{D}(\mathbf{x})}\left[\iint_{Z,Y} q_\phi(\mathbf{z}|\mathbf{x}) q_\delta(\mathbf{y}|\mathbf{z}) \log q_\delta(\mathbf{y}|\mathbf{z}) \, d\mathbf{y} d\mathbf{z}\right].$$

Thus we let the second part of our objective function be

$$\mathcal{M}_2 = \mathbb{E}_{q_\phi(\mathbf{z}|\mathbf{x})\mathcal{D}(\mathbf{x})}\left[\int_Y q_\delta(\mathbf{y}|\mathbf{z}) \log q_\delta(\mathbf{y}|\mathbf{z}) \, d\mathbf{y}\right].$$

Finally, computing $\mathcal{M}_2$ requires learning the approximate posterior $q_\delta(\mathbf{y}|\mathbf{z})$. Hence we let

$$\mathcal{N} = \mathbb{E}_{q(\mathbf{z}|\mathbf{x})\mathcal{D}(\mathbf{x}, \mathbf{y})}\left[q_\delta(\mathbf{y}|\mathbf{z})\right].$$

Thus the complete objective function consists of two parts

$$\min_{\theta,\phi,\gamma} \beta_1 \mathcal{M}_1 + \beta_2 \mathcal{M}_2$$

$$\max_\delta \beta_3 \mathcal{N}$$

where the $\beta_i$ are weights which we treat as hyperparameters. We train these parts jointly.

The terms $\mathcal{M}_2$ and $\mathcal{N}$ can be viewed as constituting an adversarial component in our model, where $q_\delta(\mathbf{y}|\mathbf{z})$ attempts to predict the label $\mathbf{y}$ given $\mathbf{z}$, and $q_\phi(\mathbf{z}|\mathbf{x})$ attempts to generate $\mathbf{z}$ which prevent this. A diagram of our CSVAE model is shown in Figure 1c.

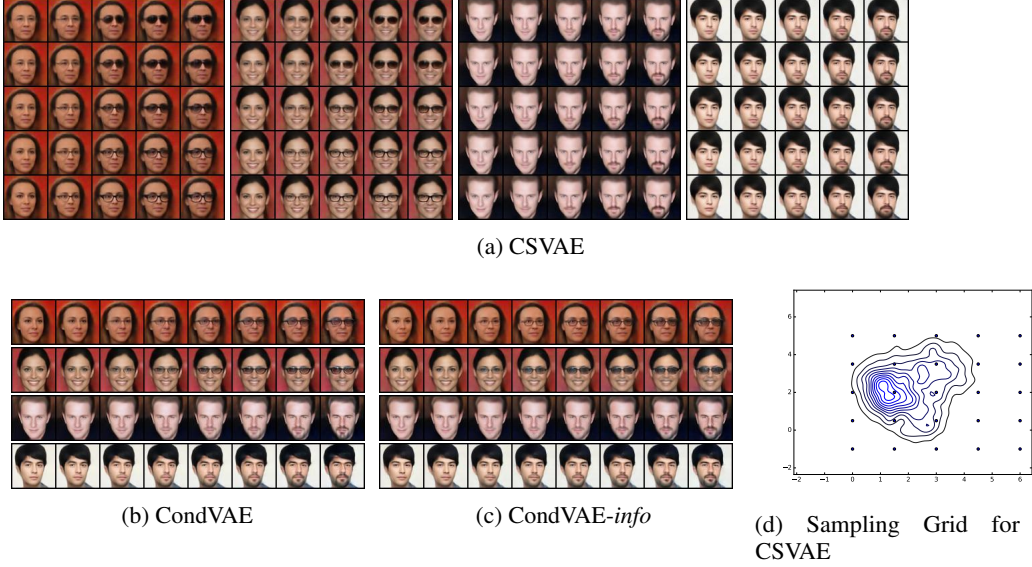

(a) CSVAE

(b) CondVAE

(c) CondVAE-*info*

(d) Sampling Grid for CSVAE

Figure 3: Images generated by each of the models when manipulating the glasses and facial hair attribute on CelebA-Glasses and CelebA-FacialHair. For CSVAE the points in the subspace $W_i$ corresponding to each image are visualized in (d) along with the posterior distribution over the test set. For CondVAE and CondVAE-*info* the points are chosen uniformly in the range $[0, 3]$. CSVAE generates a larger variety of glasses and facial hair.

## 4.2 Implementation

In practice we use Gaussian MLPs to represent distributions over relevant random variables: $q_{\phi_1}(\mathbf{z}|\mathbf{x}) = \mathcal{N}(\mathbf{z}|\mu_{\phi_1}(\mathbf{x}), \sigma_{\phi_1}(\mathbf{x}))$, $q_{\phi_2}(\mathbf{w}|\mathbf{x}, \mathbf{y}) = \mathcal{N}(\mathbf{w}|\mu_{\phi_2}(\mathbf{x}, \mathbf{y}), \sigma_{\phi_2}(\mathbf{x}, \mathbf{y}))$, and $p_\theta(\mathbf{x}|\mathbf{w}, \mathbf{z}) = \mathcal{N}(\mu_{\theta,}(\mathbf{w}, \mathbf{z}), \sigma_\theta(\mathbf{w}, \mathbf{z}))$. Furthermore $q_\delta(\mathbf{y}|\mathbf{z}) = \text{Cat}(\mathbf{y}|\pi_\delta(\mathbf{z}))$. Finally for fixed choices $\mu_1, \sigma_1, \mu_2, \sigma_2$, for each $i = 1, \ldots, k$ we let

$$p(\mathbf{w}_i|\mathbf{y}_i = 1) = \mathcal{N}(\mu_1, \sigma_1)$$
$$p(\mathbf{w}_i|\mathbf{y}_i = 0) = \mathcal{N}(\mu_2, \sigma_2).$$

These choices are arbitrary and in all of our experiments we choose $W_i = \mathbb{R}^2$ for all $i$. Hence we let $\mu_1 = (0, 0)$, $\sigma_1 = (0.1, 0.1)$ and $\mu_2 = (3, 3)$, $\sigma_2 = (1, 1)$. This implies points $\mathbf{x}$ with $\mathbf{y}_i = 1$ will be encoded away from the origin in $W_i$ and at 0 in $W_j$ for all $j \neq i$. These choices are motivated by the goal that our model should provide a way of switching an attribute on or off. Other choices are possible but we did not explore alternate priors in the current work.

It will be helpful to make the following notation. If we let $\mathbf{w}_i$ be the projection of $\mathbf{w} \in W$ onto $W_i$ then we will denote the corresponding factor of $q_{\phi_2}(\mathbf{w}|\mathbf{x}, \mathbf{y})$ as $q_{\phi_2}^i(\mathbf{w}_i|\mathbf{x}, \mathbf{y}) = \mathcal{N}(\mathbf{w}_i|\mu_{\phi_2}^i(\mathbf{x}, \mathbf{y}), \sigma_{\phi_2}^i(\mathbf{x}, \mathbf{y}))$.

## 4.3 Attribute Manipulation

We expect that the subspaces $W_i$ will encode higher dimensional information underlying the binary label $y_i$. In this sense the model gives a form of semi-supervised feature extraction.

The most immediate utility of this model is for the task of attribute manipulation in images. By setting the subspaces $W_i$ to be low-dimensional, we gain the ability to visualize the posterior for the corresponding attribute explicitly, as well as efficiently explore it and its effect on the generative distribution $p(\mathbf{x}|\mathbf{z}, \mathbf{w})$.

We now describe the method used by each of our models to change the label of $\mathbf{x} \in X$ from $i$ to $j$, by defining an attribute switching function $G_{ij}$. We refer to Section 3 for the definitions of the baseline models.

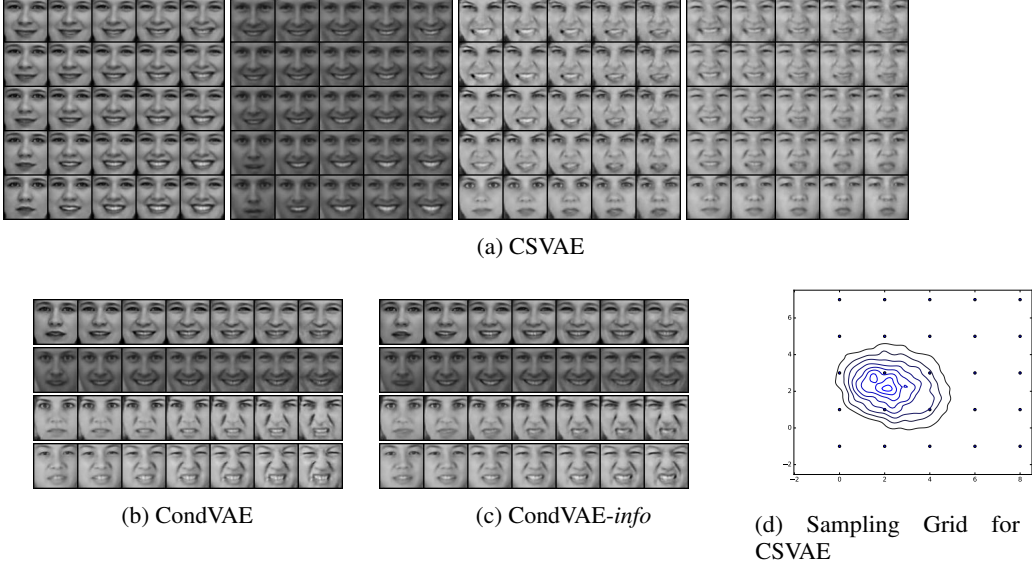

(a) CSVAE

(b) CondVAE           (c) CondVAE-*info*       (d) Sampling Grid for CSVAE

Figure 4: The analog of the results of Figure 3 on TFD for manipulating the happy and disgust expressions (a single model was used for all expressions). CSVAE again learns a larger variety of these expression than the baseline models. The remaining expressions can be seen in Figure 8.

**VAE**: For each $i = 1, \ldots, k$ let $S_i$ be the set of $(\mathbf{x}, \mathbf{y}) \in \mathcal{D}$ with $\mathbf{y}_i = 1$. Let $m_i$ be the mean of the elements of $S_i$ encoded in the latent space, that is $\mathbb{E}_{S_i} [\mu_\phi (\mathbf{x})]$. Then we define the attribute switching function

$$G_{ij} (\mathbf{x}) = \mu_\theta (\mu_\phi (\mathbf{x}) - m_i + m_j).$$

That is, we encode the data, and perform vector arithmetic in the latent space, and then decode it.

**CondVAE and CondVAE-*info***: Let $\mathbf{y}^1$ be a one-hot vector with $\mathbf{y}_j^1 = 1$. For $(\mathbf{x}, \mathbf{y}) \in \mathcal{D}$ and $p \in \mathbb{R}$ we define

$$G_{ij} (\mathbf{x}, \mathbf{y}, p) = \mu_\theta \left( \mu_\phi (\mathbf{x}, \mathbf{y}), p\mathbf{y}^1 \right).$$

That is, we encode the data using its original label, and then switch the label and decode it. We can scale the changed label to obtain varying intensities of the desired attribute.

**CSVAE**: Let $p = (p_1, \ldots, p_k) \in \prod_{i=1}^k W_i$ be any vector with $p_l = \vec{0}$ for $l \neq j$. For $(\mathbf{x}, \mathbf{y}) \in \mathcal{D}$ we define

$$G_{ij} (\mathbf{x}, p) = \mu_\theta (\mu_{\phi_1} (\mathbf{x}), p).$$

That is, we encode the data into the subspace $Z$, and select any point $p$ in $W$, then decode the concatenated vector. Since $W_i$ can be high dimensional this affords us additional freedom in attribute manipulation through the choice of $p_i \in W_i$.

In our experiments we will want to compare the values of $G_{ij} (\mathbf{x}, p)$ for many choices of $p$. We use the following two methods of searching $W$. If each $W_i$ is 2-dimensional we can generate a grid of points centered at $\mu_2$ (defined in Section 4.2). In the case when $W_i$ is higher dimensional this becomes inefficient. We can alternately compute the principal components in $W_i$ of the set $\{\mu_{\phi_2} (\mathbf{x}, \mathbf{y}) | \mathbf{y}_i = 1\}$ and generate a list of linear combinations to be used instead.

## 5 Experiments

### 5.1 Toy Data: Swiss Roll

In order to gain intuition about the CSVAE, we first train this model on the Swiss Roll, a dataset commonly used to test dimensionality reduction algorithms. This experiment will demonstrate explicitly how our model structures the latent space in a low dimensional example which can be visualized.

|  | Accuracy | | |
|---|---|---|---|
|  | TFD | CelebA-Glasses | CelebA-FacialHair |
| VAE | 19.08% | 25.03% | 49.81% |
| CondVAE | 62.97% | 96.04% | 88.93% |
| CondVAE-*info* | 62.27% | 95.16% | 88.03% |
| CSVAE (ours) | **76.23%** | **99.59%** | **97.75%** |

Table 1: Accuracy of expression and attribute classifiers on images changed by each model. CSVAE shows best performance.

We generate this data using the Scikit-learn [19] function `make_swiss_roll` with `n_samples` = 10000. We furthermore assign each data point $(x, y, z)$ the label 0 if the $x < 10$, and 1 if $x > 10$, splitting the roll in half. We train our CSVAE with $Z = \mathbb{R}^2$ and $W = \mathbb{R}^2$.

The projections of the latent space are visualized in Figure 2. The projection onto $(z_2, w_1)$ shows the whole swiss roll in familiar form embedded in latent space, while the projections onto $Z$ and $W$ show how our model encodes the data to satisfy its constraints. The data overlaps in $Z$ making it difficult for the model to determine the label of a data point from this projection alone. Conversely the data is separated in $W$ by its label, with the points labelled 1 mapping near the origin.

## 5.2 Datasets

### 5.2.1 Toronto Faces Dataset (TFD)

The Toronto Faces Dataset [23] consists of approximately 120,000 grayscale face images partially labelled with expressions (expression labels include anger, disgust, fear, happy, sad, surprise, and neutral) and identity. Since our model requires labelled data, we assigned expression labels to the unlabelled subset as follows. A classifier was trained on the labelled subset (around 4000 examples) and applied to each unlabelled point. If the classifier assigned some label at least a 0.9 probability the data point was included with that label, otherwise it was discarded. This resulted in a fully labelled dataset of approximately 60000 images (note the identity labels were not extended in this way). This data was randomly split into a train, validation, and test set in 80%/10%/10% proportions (preserving the proportions of originally labelled data in each split).

### 5.2.2 CelebA

CelebA [15] is a dataset of approximately 200,000 images of celebrity faces with 40 labelled attributes. We filter this data into two seperate datasets which focus on a particular attribute of interest. This is done for improved image quality for all the models and for faster training time. All the images are cropped as in [14] and resized to $64 \times 64$ pixels.

We prepare two main subsets of the dataset: CelebA-Glasses and CelebA-FacialHair. CelebA-Glasses contains all images labelled with the attribute $glasses$ and twice as many images without. CelebA-FacialHair contains all images labelled with at least one of the attributes $beard$, $mustache$, $goatee$ and an equal number of images without. Each version of the dataset therefore contains a single binary label denoting the presence or absence of the corresponding attribute. This dataset construction procedure is applied independently to each of the training, validation and test split.

We additionally create a third subset called CelebA-GlassesFacialHair which contains the images from the previous two subsets along with the binary labels for both attributes. Thus it is a dataset with multiple binary labels, but unlike in the TFD dataset these labels are not mutually exclusive.

## 5.3 Qualitative Evaluation

On each dataset we compare four models. A standard VAE, a conditional VAE (denoted here by CondVAE), a conditional VAE with information factorization (denoted here by CondVAE-*info*) and our model (denoted CSVAE). We refer to Section 3 for the precise definitions of the baseline models.

We examine generated images under several style-transfer settings. We consider both *attribute transfer*, in which the goal is to transfer a specific style of an attribute to the generated image, and *identity transfer*, where the goal is to transfer the style of a specific image onto an image with a different identity.

Figure 3 shows the result of manipulating the glasses and facial hair attribute for a fixed subject using each model, following the procedure described in Section 4.3. CSVAE can generate a larger variety of both attributes than the baseline models. On CelebA-Glasses we see a variety of rims and different styles of sunglasses. On CelebA-FacialHair we see both mustaches and beards of varying thickness. Figure 4 shows the analogous experiment on the TFD data. CSVAE can generate a larger variety of smiles, in particular teeth showing or not showing, and open mouth or closed mouth, and similarly for the disgust expression.

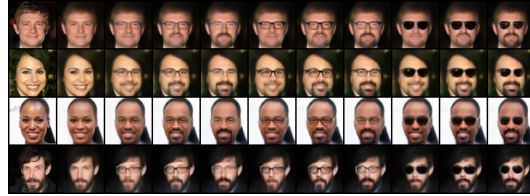

Figure 5: Attribute transfer with a CSVAE on CelebA-GlassesFacialHair. From left to right: input image, reconstruction, Cartesian product of three representative glasses styles and facial hair styles. Additional attribute transfer results are provided in the supplementary material.

We also train a CSVAE on the joint CelebA-GlassesFacialHair dataset to show that it can independently manipulate attributes as above in the case where binary attribute labels are not mutually exclusive. The results are shown in Figure 5. Thus it can learn a variety of styles as before, and manipulate them simultaneously in a single image.

Figure 6 shows the CSVAE model is capable of preserving the style of the given attribute over many identities, demonstrating that information about the given attribute is in fact disentangled from the $Z$ subspace.

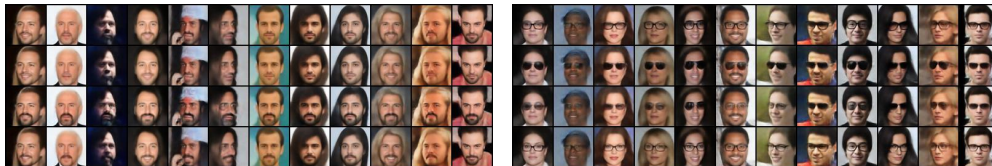

Figure 6: Style transfer of facial hair and glasses across many identities using CSVAE.

## 5.4 Quantitative Evaluation

**Method 1**: We train a classifier $C : X \longrightarrow \{1, \ldots, K\}$ which predicts the label $\mathbf{y}$ from $\mathbf{x}$ for $(\mathbf{x}, \mathbf{y}) \in \mathcal{D}$ and evaluate its accuracy on data points with attributes changed using the model as described in Section 4.3.

A shortcoming of this evaluation method is that it does not penalize images $G_{ij}(\mathbf{x}, \mathbf{y}, p_j)$ which have large negative loglikelihood under the model, or are qualitatively poor, as long as the classifier can detect the desired attribute. For example setting $p_j$ to be very large will increase the accuracy of $C$ long after the generated images have decreased drastically in quality. Hence we follow the standard practice used in the literature, of setting $p_j = 1$ for the models CondVAE and CondVAE-*info* and set $p_j$ to the empirical mean $\mathbb{E}_{S_j}\left[\mu_{\phi_2}^j(\mathbf{x})\right]$ over the validation set for CSVAE in analogy with the other models. Even when we do not utilize the full expressive power of our model, CSVAE show better performance.

Table 1 shows the results of this evaluation on each dataset. CSVAE obtains a higher classification accuracy than the other models. Interestingly there is not much performance difference between CondVAE and CondVAE-*info*, showing that the information factorization loss on its own does not improve model performance much.

**Method 2** We apply this method to the TFD dataset, which comes with a subset labelled with identities. For a fixed identity $t$ let $S_{i,t} \subset S_i$ be the subset of the data with attribute label $i$ and identity $t$. Then

|  | target - changed | original - changed | target - original |
|---|---|---|---|
| VAE | 75.8922 | 13.4122 | 91.2093 |
| CondVAE | 74.3354 | 18.3365 | 91.2093 |
| CondVAE-*info* | 74.3340 | 18.7964 | 91.2093 |
| CSVAE (ours) | **71.0858** | 28.1997 | 91.2093 |

Table 2: MSE between ground truth image and image changed by model for each subject and expression. CSVAE exhibits the largest change from the original while getting closest to the ground truth.

over all attribute label pairs $i, j$ with $i \neq j$ and identities $t$ we compute the mean-squared error

$$L_1(p) = \sum_{i,j,t,i \neq j} \sum_{x_1 \in S_{i,t}, x_2 \in S_{j,t}} (x_2 - G_{ij}(\mathbf{x}_1, \mathbf{y}_1, p_j))^2. \tag{6}$$

In this case for each model we choose the points $p_j$ which minimize this loss over the validation set.

The value of $L_1$ is shown in Table 2. CSVAE shows a large improvement relative to that of CondVAE and CondVAE-*info* over VAE. At the same time it makes the largest change to the original image.

## 6  Conclusion

We have proposed the CSVAE model as a deep generative model to capture intra-class variation using a latent subspace associated with each class. We demonstrated through qualitative experiments on TFD and CelebA that our model successfully captures a range of variations associated with each class. We also showed through quantitative evaluation that our model is able to more faithfully perform attribute transfer than baseline models. In future work, we plan to extend this model to the semi-supervised setting, in which some of the attribute labels are missing.

**Acknowledgements**

We would like to thank Sageev Oore for helpful discussions. This research was supported by Samsung and the Natural Sciences and Engineering Research Council of Canada.

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
