[Supplementary Material]

# 7  Appendix

## 7.1  Architectures and optimization

We implement our models in PyTorch [18]. We use the same architectures and hyperparameters in all our experiments.

We train our models for 2300 epochs using Adam optimizer with $betas = (0.9, 0.999)$, $eps = 10^{-8}$ and initial $lr = 10^{-3}/2$. We use PyTorch's learning rate scheduler MultiStepLR with $milestones = \left\{ 3^i \mid i = 0, \ldots, 6 \right\}$ and $gamma = 0.1^{1/7}$. We use minibatches of size 64.

Our architectures consist of convolutional layers with ReLu activations which roughly follow that found in [14].

Our loss function is weighted as

$$- \beta_1 \mathbb{E}_{q_\phi(\mathbf{z}, \mathbf{w} | \mathbf{x}, \mathbf{y})} \left[ \log p_\theta \left( \mathbf{x} \mid \mathbf{w}, \mathbf{z} \right) \right] + \beta_2 D_{KL} \left( q_\phi \left( \mathbf{w} \mid \mathbf{x}, \mathbf{y} \right) \parallel \log p \left( \mathbf{w} \mid \mathbf{y} \right) \right)$$

$$+ \beta_3 D_{KL} \left( q_\phi \left( \mathbf{z} \mid \mathbf{x}, \mathbf{y} \right) \parallel p \left( \mathbf{z} \right) \right) + \beta_4 \mathbb{E}_{q_\phi(\mathbf{z} | \mathbf{x}) \mathcal{D}(\mathbf{x})} \left[ \int_Y q_\delta \left( \mathbf{y} \mid \mathbf{z} \right) \log q_\delta \left( \mathbf{y} \mid \mathbf{z} \right) d\mathbf{y} \right]$$

$$- \log p \left( \mathbf{y} \right)$$

$$\beta_5 \mathbb{E}_{q(\mathbf{z} | \mathbf{x}) \mathcal{D}(\mathbf{x}, \mathbf{y})} \left[ \log q_\delta \left( \mathbf{y} \mid \mathbf{z} \right) \right].$$

We use the values $\{\beta_1 = 20, \beta_2 = 1, \beta_3 = 0.2, \beta_4 = 10, \beta_5 = 1\}$.

Our hyperparameters were determined by a grid search using both quantitative and qualitative analysis (see below) of models trained for 100,300, and 500 epochs on a validation set. Stopping time was determined similarly.

## 7.2  Additional results

|  | anger | disgust | fear | happy | sad | surprise | neutral | final |
|---|---|---|---|---|---|---|---|---|
| VAE | 12.61% | 7.72% | 2.50% | 30.24% | 5.65% | 6.25% | 68.57% | 19.08% |
| CondVAE | 58.92% | 66.98% | 34.95% | 91.19% | 43.39% | 53.36% | 91.97% | 62.97% |
| CondVAE-*info* | 57.64% | 64.79% | 32.76% | 92.68% | 43.69% | 52.36% | 91.95% | 62.27% |
| CSVAE | **79.04%** | **85.11%** | **53.50%** | **98.70%** | **47.09%** | **71.49%** | **98.70%** | **76.23%** |

Table 3: Accuracy of an expression classifier on images changed by each model. CSVAE shows best performance.

|  | CelebA-Glasses | | | CelebA-FacialHair | | |
|---|---|---|---|---|---|---|
|  | Glasses | Neutral | Final | Facial Hair | Neutral | Final |
| VAE | 5.04% | 65.01% | 25.03% | 38.46% | 61.17% | 49.81% |
| CondVAE | **100.00%** | 88.13% | 96.04% | **100.00%** | 77.86% | 88.93% |
| CondVAE-*info* | **100.00%** | 85.49% | 95.16% | 99.97% | 76.10% | 88.03% |
| CSVAE | 99.38% | **100.00%** | **99.59%** | **100.00%** | **95.50%** | **97.75%** |

Table 4: Classifier accuracy on the CelebA-Glasses (left) and CelebA-FacialHair (right) datasets when performing attribute transfer.

Figure 7: Additional attribute transfer results with a CSVAE trained on CelebA-GlassesFacialHair. From left to right: input image, reconstruction, Cartesian product of three representative glasses styles and facial hair styles.

(a) Happiness

(b) Disgust

(c) Fear

(d) Sadness

(e) Surprise

(f) Anger

Figure 8: More results of the experiment presented in Figure 4 on TFD. We demonstrate manipulating each of the expressions in the dataset. The first three expressions display more 2-dimensional variation than the last three. This is likely due to the content of the dataset. A single model was used for all images.

Figure 9: More results of the experiment presented in Figure 3 on a dataset with the heavy makeup attribute.

Figure 10: Comparisons of different models changing the expression of a face. The columns are left to right: VAE, CondVAE, CSVAE. The first row is the original, the second is a reconstruction. Each subsequent row is a different expression generated by the model.

Figure 11: The distribution over $W_i$ output by the model on the test set for each expression $i$ in the order 0 = Anger, 1 = Disgust, 2 = Fear, 3 = Happy, 4 = Sad, 5 = Surprise.