[Reviews · NeurIPS 2018]

Reviewer 1



Updated (due to rebuttal & discussion w/ R2): The authors reiterate in their rebuttal their core contributions of "extracting information beyond binary labels" and "attribute manipulation from a single image", together with the promise to clarify it in the paper. The contributions are relevant to the community, since this form of hierarchical disentangling seems novel. That said, there is some degree of similarity of the proposed variational approach to IGN (Deep Convolutional Inverse Graphics Network https://arxiv.org/abs/1503.03167). IGN is cited, but not discussed in detail, and an empirical comparison is not provided, despite being applicable to the current setting as well. Nevertheless, since the selling point of the paper seems to be the ability to discover sub-categories from only category labels, which is not addressed in IGN and is an interesting empirical find, I increased my score to be marginally above the acceptance threshold. --- The paper proposes an approach for learning variational auto-encoders for labeled data in which the latent space consists of two parts: one that captures the labels information and one that captures the remaining information for producing the data. Along with the variational lower-bound corresponding to the introduced latent variables, the proposed training objective additionally penalizes the mutual information between the latent variable corresponding to the remaining data and the labels. The approach is illustrated on a toy example and two image datasets, the Toronto Faces dataset and CelebA. I think the approach is described clearly, and several experiments, qualitative and quantitative, are presented. That said, since there are many works on the topic, some seemingly quite similar in the use of variational auto-encoders with various regularizers for disentangling latent representations, I worry that the proposed approach only incrementally contributes to the topic. See below for specific questions on this matter. Comments: * The work on building fair representations (where the learned representations are encouraged to be independent of "sensitive" factors) seems relevant and not discussed. Could you highlight the distinction? For instance, The Variational Fair Autoencoder [1] seems relevant, where the additional "fairness" regularizer uses maximum mean discrepancy rather than mutual information. From this perspective, I believe that the contribution of the current paper is the use of the mutual information regularizer, and a discussion on the benefits of one metric versus the other would be useful. From a quick search, the idea of using mutual information to "obfuscate" sensitive information is also mentioned in [3], though in a somewhat different setting. * In the related work section, I wish the authors provided more detail of the proposed approach with respect to the cited works (e.g. w.r.t. [21-23] and w.r.t. [3]) so that we can understand which specific contributions are being made and how to evaluate their merits. For instance, the fact that previous work used GANs rather than VAEs (lines 61-62) is a pretty weak point in favor of the paper, unless properties of VAEs are shown to be necessary for solving the task, or the method shown to empirically perform much better than the GAN counterparts. Experiments: * I like the toy example, it's illustrative and a good sanity check for the method. * None of the presented applications is focused on the specific contributions of this paper, e.g. the additional mutual information regularizer. Instead, the experiments are quite generic for attribute manipulation/style transfer which have been presented in many works recently, e.g. the ones mentioned in the related work. From this perspective, the empirical contribution does not stand out. [1] https://arxiv.org/abs/1511.00830 ICLR 2016 [2] https://www.cs.toronto.edu/~toni/Papers/icml-final.pdf ICML 2013 Summary: The submission is ok, but, unless the contributions are highlighted more clearly, it only incrementally contributes to the literature, hence the slightly negative score.

Reviewer 2



Summary of main ideas of the submission The authors propose an extension of VAEs to a setting in which unsupervised latent subspaces are discovered. The latent subspaces capture meaningful attributes of each class. Relation to previous work The proposed CVAE is a straightforward extension of the semi-supervised VAE of Kingma et al. This relationship is clearly explained, a more detailed discussion of subspace analysis is missing. Quality The work is thoroughly executed and all derivations are sound. The experiments are well-done, but fail to clearly highlight the selling point of the method. Disentanglement of attributes has been achieved in various papers and it is unclear how this particular model advances the state of the art, or even how it exactly compares to it. Clarity The paper is clearly written. The motivation is not very clear and the sizes of subspaces are chosen heuristically with no clear explanation. It would have also been helpful if the authors had included a graphical model diagram or at least a figure that displays the exact architecture used in some of the experiments. It is also not clear to me how attributes can be shared among classes and why it would be desirable to represent classes in independent, non-overlapping subspaces. Originality The work is original, as only a few works have added additional unsupervised structure into the latent space. However, the originality is limited by the fact, that there is no clear explanation as to why the proposed structure is superior to other approaches that disentangle attributes from identity. Significance The paper is not very significant, as no new application or necessity for the work is presented. No urgent point as to why this work should be presented at a top-tier conference like NIPS is brought across. ======= Update Rebuttal: I found the paper interesting and the results relevant. However, the presentation was not clear enough, especially with respect to how the main contributions have been presented. After reading the restated core contributions in the rebuttal and reading through the paper several times again, I found the core contribution (being able to manipulate class-specific attributes from a single example) much clearer. This is indeed a relevant and interesting novelty. It would be great if the authors would have shown the proposed algorithm visually and put a stronger focus on the implications of their core contributions. Especially the role of the unstructured z space would have been much easier to grasp with a figure of the graphical model. I like the paper better quite a bit better after reconsideration and thank the authors for the insightful rebuttal. As the authors address the main points of criticism and promise to rewrite the paper to make all of this more clear, I raised my score.

Reviewer 3



The paper proposes a variant of a VAE model for labeled datasets. The model explicitly encodes the information corresponding to the label in a predetermined set of components of the latent code. Experimental evaluation on facial images are presented, showing that the model is able to capture the intra-class variability linked to a given class and performa tribute transfer. The paper is well written and easy to follow. A few references are missing in my opinion (please see bellow). The idea of the paper is very interesting. The experimental evaluation seems convincing but limited. I believe that the paper would be stronger if more examples where presented (not only facial images). An very interesting example could be to have different dog breads as classes. The model assigns a low dimensional subspace for each of the classes that the given label can take. This model is clear when there is little information shared among classes, such as in the binary labels (e.g. with or without sunglasses). But when this is not the case, this might not be the best way of factorizing the latent space (the number of variables grows linearly with the number of classes). An extreme example would be to use as labels the identity of the person in the picture. In the case of the facial expressions, It would make sense for some of the latent variables to be shared among the different classes. As the labels seem a discretization of a continuous space, and an complete factorization. In Figure 3, what happens if one changes the value of some of the variables corresponding to the other classes while keeping the class of interest fixed? It would be interesting to consider more than one label at the same time, as normally there are many factors that explain the data (as in the CelebA dataset). For instance to consider both facial hair as well as the presence of glasses (and two variables y_1 and y_2). How do you think that the model would scale up? The authors should cite the variational fair auto encoder [A]. The authors propose a VAE that factorizes the latent variables from a given sensitive variable (e.g. gender). In order to reduce dependence of the representation on the sensitive variable they use MMD as an alternative to using minimizing the mutual information. The idea of using an adversary to minimize the dependence of the representation with a given variable has also been explored in [B] and [C]. While the formulation is different, the work [D] seems a relevant reference. [A] Louizos, Christos, et al. "The variational fair autoencoder." ICLR 2016 [B] Edwards, Harrison, and Amos Storkey. "Censoring representations with an adversary." ICLR 2016 [C] Ganin, Yaroslav, et al. "Domain-adversarial training of neural networks." JMLR 2016 [D] Mathieu, M, et al. "Disentangling factors of variation in deep representation using adversarial training." NIPS. 2016.